# Design of New Primer Sets for the Development of a Loop-Mediated Isothermal Amplification for Rapid Detection of *Neisseria meningitidis*

**DOI:** 10.3390/cimb47060467

**Published:** 2025-06-17

**Authors:** Cuong Viet Vo, Trang Thu Nguyen, Huong Thu Ngo, Lan Anh Thi Bui, Toan Van Trinh, Loan Thi Vu, Hieu Dang Hoang, Phong Quoc Truong

**Affiliations:** 1Joint Vietnam-Russian Tropical Science and Technology Research Center, Hanoi 100000, Vietnam; cuongvrtc@gmail.com (C.V.V.); lananhrus@gmail.com (L.A.T.B.); tvtoan210594@gmail.com (T.V.T.); vtloan194@gmail.com (L.T.V.); hoang.dang.hieu.ls@gmail.com (H.D.H.); 2Institute of Health Science and Technology, Hanoi University of Science and Technology, Hanoi 100000, Vietnam; nguyenthutrang16198@gmail.com; 3School of Chemistry and Life Sciences, Hanoi University of Science and Technology, Hanoi 100000, Vietnam; 4Center for Research and Production of Vaccines and Biologicals, Ministry of Health, Hanoi 100000, Vietnam; huongngothu79@gmail.com

**Keywords:** loop-mediated isothermal amplification (LAMP), *Neisseria meningitidis*, new primer set, optimization, rapid detection, visual inspection

## Abstract

*Neisseria meningitidis* is a pathogenic bacterial agent that causes meningococcal meningitis in humans. Developing a rapid and low-cost *N. meningitidis* detection method is crucial, especially for developing countries. This study focuses on the development of an efficient loop-mediated isothermal amplification (LAMP) method for accurate *N. meningitidis* identification. A new LAMP primer set was designed, and a LAMP reaction was optimized. The colorimetric detection method was also applied, and the assay characteristics were evaluated using clinical samples. The results demonstrated a specific LAMP reaction for *N. meningitidis* detection of genotypes A, B, and C, with a limit of detection of 10^2^ cfu/mL, 100% specificity and sensitivity, and a rapid detection time of only 40 min by colorimetric visual inspection. No cross-reactivity with reference strains of *Streptococcus pneumoniae*, *Staphylococcus aureus*, *Neisseria lactamica*, *Mycobacterium tuberculosis*, and *Haemophilus influenzae* type b was observed in the LAMP reaction with the new primer set. This result suggests that the LAMP reaction could be a promising tool for developing a rapid *N. meningitidis* detection method suitable for use in Vietnam and other developing countries.

## 1. Introduction

*Neisseria meningitidis*, a pathogenic member of the genus *Neisseria*, is a Gram-negative diplococcus bacterium that causes septicemia and meningitis in humans [1]. With serious symptoms such as high fever, headache, vomiting, and long-term consequences even with good treatment, meningococcal disease caused by *N. meningitidis* is a major health threat. The capsular polysaccharide is a main virulence factor of *N. meningitidis*, and its structural differences are used to classify the bacterium into 12 serogroups. Among these serogroups, A, B, C, W, X, and Y are the most commonly found worldwide [2]. The highest infection rate is observed in Africa, especially in sub-Saharan Africa, with serogroup A as a leading type, followed by serogroups W and X [3]. Serogroups B, C, and Y are mainly detected in Europe, the United States, and Canada [4]. In the Asia–Pacific serogroups, B, W, and Y dominate, although some cases are caused by serogroups A and C [5].

In Vietnam, the latest surveillance data reveals that meningococcal meningitis caused by *N. meningitidis* is a dangerous disease with a high mortality rate, and children under 5 years old are the most vulnerable group [6]. Another study estimated a high incidence rate of the disease among soldiers in the Vietnam People’s Army, with the highest rate of 3.3/100,000 soldiers in 2016 [7]. Most meningococcal cases in Vietnam are caused by serogroup B; however, the specific information on pathogenicity for other serogroups of *N. meningitidis* is limited and needs further studies [6].

The gold standard for *N. meningitidis* diagnosis is bacterial culture from cerebrospinal fluid (CSF) and peripheral blood samples on special media, such as blood agar or chocolate agar plates [8]. However, a false-negative result may occur if the samples have been previously treated with antibiotics or if the culturing method is not accurate. Furthermore, the bacterial isolation technique requires a well-equipped laboratory with highly skilled technicians [9,10]. Another common method for *N. meningitidis* detection is latex agglutination; however, the timing of specimen collection is an issue and has been associated with false negatives and low sensitivity [11].

To overcome the limitations of the traditional methods, several nucleic acid tests (NATs), including PCR, real-time PCR, and qPCR, have been developed. Some PCR-based methods for detecting *N. meningitidis* by amplifying the polysialyltransferase *siaD* gene or other genes involved in capsule synthesis have been reported [12,13], as well as real-time and multiplex PCR assays [14,15]. However, PCR assays need special equipment, and these methods are too complicated to perform for point-of-care testing. The loop-mediated isothermal amplification, or LAMP, is a suitable option compared to PCR. Unlike PCR, the LAMP reaction utilizes a DNA polymerase with a strand displacement activity and a set of four specific primers that specifically target six different regions on the DNA template.

The significant advantage of LAMP is that this method operates without the need for complex instrumentation and can amplify specific DNA in a short time [16]. While the design of LAMP primers is more complex due to the requirement for three primer sets (inner primers FIP/BIP, outer primer F3/B3, and loop primer FL/BL), this minor drawback can be overcome by the development of bioinformatics tools.

Some LAMP methods for *N. meningitidis* detection have been reported up to now. In 2011, McKenna et al. were the first group to develop a LAMP reaction for *N. meningitidis* detection [17]. In 2015, Do Kyung Lee et al. designed a LAMP reaction that can successfully detect *N. meningitidis* in cerebrospinal fluid samples (CFS) [18,19]. In 2018, Higgins et al. established a TEC-LAMP reaction to identify multi-bacterial meningitis pathogens [20]. However, there are currently no commercially available LAMP kits for detecting *N. meningitidis*; therefore, this area of research still requires further investigation.

Recently, we successfully designed a LAMP method to specifically and rapidly detect *N. meningitidis* with the *ctrA* gene as a target. The coverage, sensitivity, and specificity of the LAMP reactions were evaluated with clinical samples. We propose that the LAMP reaction developed in this study could serve as an effective and suitable diagnostic method for *N. meningitidis* detection in Vietnam as well as in other developing countries.

## 2. Materials and Methods

### 2.1. Materials

*Bst* DNA polymerase and dNTPs were purchased from New England Biolabs (NEB, Ipswich, MA, USA). LAMP primers were custom-synthesized from Integrated DNA Technologies (Coralville, IA, USA). The i-Star Taq^TM^ DNA polymerase kit was purchased from iNtRON Biotechnology (Seongnam-si, Gyeonggi-do, Korea). DNA ladder 100 bp was obtained from Thermo Scientific (Waltham, MA, USA).

Purified DNA templates of *Neisseria meningitidis* were provided by the Joint Vietnam–Russian Tropical Science and Technology Research Center (Hanoi, Vietnam).

Thirty positive samples from patients diagnosed with meningococcal meningitis and 30 negative samples from patients diagnosed as not having meningococcal meningitis but with symptoms like those of patients with the disease were provided by the Joint Vietnam-Russian Tropical Science and Technology Research Center (Hanoi, Vietnam). All samples were purified DNA templates and had been previously confirmed by PCR for the presence or absence of *N. meningitidis*.

### 2.2. Primer Design

The *ctrA* gene sequences of *Neisseria meningitidis* from several serogroups were taken from NCBI at https://www.ncbi.nlm.nih.gov/ (accessed on 10 July 2022). The Multalin bio tool at http://multalin.toulouse.inra.fr/multalin/ (accessed on 10 July 2022) was used to identify the conserved region of the *ctrA* genome.

Four LAMP primer sets were examined. The first primer set was based on the publication of McKenna et al. [17]. The second primer set was based on the first with three modifications to primers FIP, FL, and F3. The remaining two primer sets were designed based on the conserved region of the *ctrA* genome using Primer Explorer V5 online LAMP primer design software at https://primerexplorer.eiken.co.jp/ (accessed on 10 July 2022).

### 2.3. PCR Amplification of Target Gene

Based on the selected primer set, the PCR reaction was performed with 3 DNA templates from 3 serogroups, A, B, and C, to amplify a portion of the *crtA* gene sequence as a template for the LAMP reaction. Briefly, the PCR reaction was mixed in 20 µL, containing 2 µL 10X buffer, 2 µL dNTPs (10 mM), 0.8 µL for each FIP (5 µM), and BIP (5 µM), 0.6 µL of i-Star Taq^TM^ DNA polymerase, 1 µL of purified DNA template, and sterile double distilled water (ddH_2_O). The PCR cycle included the following: initial denaturation at 95 °C for 2 min; 35 cycles of 95 °C for 30 s, 57.5 °C for 30 s, and 72 °C for 1 min 30 s; and final extension at 72 °C for 5 min. After amplification, the PCR products were separated on a 0.8% agarose gel with the DNA ladder 100 bp from New England Biolabs (NEB, Ipswich, MA, USA) as a marker.

### 2.4. LAMP Reaction and Optimization

The LAMP reaction was performed based on the publication of Notomi et al. [16], with some modifications. Briefly, the LAMP preliminary reaction was mixed in 25 µL containing 10X buffer, 2 µM for each FIB/BIP, 0.2 µM for each F3/B3, 0.2 µM for each FL/BL, 8 mM of MgSO_4_, 1.5 mM of dNTPs, 0.8 M of betaine, and 8 IU *Bst* DNA polymerase. The reaction was heated at 65 °C for 60 min, and then the LAMP products were separated into 2% agarose gel electrophoresis. The start copy number of the DNA template was also tested before optimizing other conditions, with five values being 10^3^, 5 × 10^3^, 10^4^, 5 × 10^4^, and 10^5^, respectively.

Based on the preliminary reaction setup, other conditions were optimized, including the concentration of FIP/BIP primers (0.25–2 µM); the concentration of FL/BL primers (0 µM–1.2 µM); the concentration of dNTPs (0.1 mM–1.5 mM); the concentration of MgSO_4_ (4 mM–12 mM), the total amount of *Bst* DNA polymerase (2 IU–10 IU); the concentration of betaine (0 M–1.2 M); the melting temperature (60 °C–65 °C); the reaction time (10 min–60 min); and the detection threshold (10^0^–10^6^ CFU/mL).

### 2.5. Visual Detection of LAMP Product Using pH-Sensitive Dye

The visual detection method for LAMP product evaluation was performed following Tanner et al. [21]. In brief, the color change of the indicator was assessed under various buffer conditions with different pH values. Then, the optimum LAMP reaction was carried out using the pH-sensitive dye in a proportion appropriate for the DNA template. Finally, the LAMP products were evaluated by the naked eye.

### 2.6. Sensitivity and Specificity of the LAMP Reaction

The LAMP reaction was performed with five clinical samples belonging to three main serogroups, A, B, and C, to evaluate the coverage of the selected LAMP primer set. The LAMP reaction was performed with thirty positive samples from patients diagnosed with meningococcal meningitis for the sensitivity evaluation. The LAMP reaction was performed with thirty negative samples from patients diagnosed as not having meningococcal meningitis but with symptoms like those of patients with the disease for the specificity evaluation.

### 2.7. Statistical Analysis

Sensitivity and specificity for the LAMP reaction were calculated using the following formulas:Sensitivity (%) = [True positive/(True positive + False Negative)] × 100Specificity (%) = [True negative/(True negative + False Positive)] × 100

The calculation of the lower and upper limits of the confidence interval for a proportion was performed using a tool available at http://vassarstats.net/ (accessed on 1 January 2023).

## 3. Results

### 3.1. Designing of New LAMP Primers for Detecting N. meningitidis

In the LAMP reaction, primers are critical components that determine the sensitivity and specificity of the assay. Aiming to design stable and effective primers, four different LAMP primer sets were used to perform the LAMP reaction in this study (Table 1). The first primer set was based on the publication of McKenna et al. [17]. The second primer set was based on the first one, with modifications in the FIP, FL, and F3 primers. The third and fourth primer sets were designed based on the conserved region (1) and the conserved region (2) of the *ctrA* gene (Table 2), respectively.

LAMP reactions were performed to select the specific LAMP primer set for detecting *Neisseria meningitidis* among four primer sets. Each primer set was tested with one negative and one positive sample. The gel electrophoresis results showed three primer sets, (1), (2), and (3), witnessed false-positive results during the first LAMP reaction attempt, while primer set (4) did not show this phenomenon (Figure 1).

For primer set (4), the F2 region of the FIP primer and the B2 region of the BIP primer were different from what was published in [17] (Figure 2). Previous study by Lee et al. (2015) [18] also showed that when mutating this FIP primer, only one out of three primers gave specific results. Therefore, the primer set (4) was chosen for further study.

### 3.2. Optmization of LAMP Reactions

#### 3.2.1. Concentration of FIP/BIP Primers

In the LAMP reaction, the primers of FIP/BIP play a key role in creating the initiation of the LAMP reaction. The concentration ranges of the FIP/BIP primers of 0.25 µM, 0.5 µM, 1 µM, 1.5 µM, and 2 µM were investigated. The results showed that a positive signal appeared at all concentrations of FIP/BIP primers; however, the signal intensity was proportional to the increase in primer concentration and reached a saturated status of 1.5 µM (Figure 3). Required concentration of FIP/BIP primers in this study is lower than in previous studies [17,18]. From the obtained results, the concentration of FIP/BIP primers of 1.5 µM will be selected for further experiments to achieve high sensitivity.

#### 3.2.2. Concentration of FL/BL Primers

The loop primers do not directly initiate the elongation reaction to synthesize new DNA strands, but they play a very important role in the LAMP reaction. The loop primer has the role of attaching to the loop region of the LAMP product to create favorable conditions for the FIP and BIP primers to attach to complementary positions on this loop region, thereby enhancing DNA synthesis efficiency [22]. A range of FL/BL primer concentrations from 0 µM to 0.1 µM, 0.2 µM, 0.4 µM, 0.8 µM, and 1.2 µM was investigated. The obtained result showed that no LAMP reaction occurred without loop primers. However, the LAMP signal was observed on the agarose gel electrophoretic pattern when adding loop primers (Figure 4). The signal intensity was increased with a higher concentration of primers and reached a saturated signal intensity at 0.8 µM of each loop primer. No LAMP reaction product without loop primers at a low level of template was observed in the previous study [23]. This result indicated that LAMP reactions with designed primers in this study were accelerated by using loop primers. A suitable concentration of loop primers of 0.8 µM was chosen for further experiments.

#### 3.2.3. Concentration of dNTPs

In the LAMP reaction, the concentration of dNTPs is usually in the range of 0.4 mM–1.4 mM. In this study, the dNTPs concentration ranges of 0.1 mM, 0.5 mM, 1.0 mM, and 1.5 mM were investigated. The LAMP signal was only observed at the dNTPs concentrations of 1.0 and 1.5 mM, while no LAMP signal was observed at the lower concentrations of dNTPs (Figure 5). The obtained results are consistent because the LAMP reaction requires a higher amount of dNTPs than the conventional PCR reaction due to the production of a large number of amplicons.

#### 3.2.4. Concentration of MgSO_4_

MgSO_4_ is one of the factors that plays a very important role in the PCR reaction. The Mg^2+^ ion acts as a cofactor of DNA polymerase enzymes by allowing the binding of dNTPs during the reaction. Mg^2+^ ions at the active site of the enzyme catalyze the formation of a phosphodiester bond between the 3′-OH of the primer and the phosphate group of a dNTP. In addition, Mg^2+^ facilitates the formation of complexes between primers and DNA templates by stabilizing the negative charge on their phosphate backbone. Therefore, optimizing the Mg^2+^ concentration in the LAMP reaction plays a necessary role in terms of specificity and reaction efficiency. In a PCR reaction, the appropriate MgSO_4_ concentration is usually 2 mM. However, in the LAMP reaction, the amount of product produced is very large. Therefore, the Mg^2+^ concentration needs to be increased to suit the reaction. In this study, a range of MgSO_4_ concentrations of 4 mM, 6 mM, 8 mM, 10 mM, and 12 mM was investigated. As a result, at concentrations of 10 mM and 12 mM, no product of the LAMP reaction appeared. This can be explained by the fact that, at these two concentrations, the amount of Mg^2+^ is too high, leading to the inhibition of the activity of the Bst DNA polymerase enzyme. The signal band intensity at 8 mM is lighter than at 4 mM and 6 mM. At concentrations of 4 mM and 6 mM, the signal intensities of the bands are quite similar (Figure 6). Therefore, a MgSO_4_ concentration of 4 mM was considered for further experiments.

#### 3.2.5. Concentration of Betaine

Betaine is a surfactant used in PCR reactions to inhibit the secondary structures of DNA templates or DNA primers. In addition, it also reduces interference reactions and enhances the separation of GC-rich DNA strands. In the LAMP reaction, betaine is usually added at a concentration of 0.5–1.6 M. However, betaine, when used in high concentrations, can reduce the activity of DNA polymerase enzymes, leading to reduced reaction efficiency. A range of betaine concentrations of 0 M, 0.4 M, 0.8 M, and 1.2 M was investigated. The result showed that the LAMP reaction occurred at different investigated concentrations of betaine. The lowest signal intensity of the product was observed without the addition of betaine in the reaction. The signal intensity of the product tended to increase with increasing betaine concentration (Figure 7). The highest concentration of 1.2 M did not affect the efficiency of the LAMP reaction in this study.

#### 3.2.6. Amount of Bst DNA Polymerase

Bst DNA polymerase is a heat-stable enzyme at an elevated temperature of 65 °C. This enzyme can synthesize DNA at a constant temperature, and the synthesis process is not inhibited by the secondary structure of DNA. For these reasons, Bst DNA polymerase is included in the LAMP reaction to amplify the DNA template. In theory, the efficiency of the reaction is dependent on the amount of enzyme used. A lower amount of enzyme leads to a decrease in the reaction rate. In contrast, a higher amount of enzyme increases the reaction efficiency. However, if the concentration of enzyme is too high, it may inhibit the reaction due to the competition of enzymes with substrate. In addition, excess use of enzymes can cause waste. An optimal amount of Bst DNA polymerase for the LAMP reaction was determined from a range of 2, 4, 6, 8, and 10 IU. The result showed that the signal intensity of the product was proportional to the amount of enzyme used in the range of 2–6 IU. The amount of product did not increase with a higher amount of enzyme used in a reaction (6–10 IU) (Figure 8).

#### 3.2.7. Reaction Temperature

According to the properties of Bst DNA polymerase, this enzyme works effectively in the temperature range of 60–65 °C. The LAMP reaction is performed at a stable temperature condition. This temperature condition must ensure optimal enzyme activity and the highest primer-binding efficiency. Therefore, the reaction temperature needs to be optimized for each certain primer set. A range of reaction temperatures of 61–65 °C was applied to the LAMP reaction to amplify the target gene. A similar signal intensity of the LAMP product was observed at all reaction temperatures (Figure 9).

#### 3.2.8. Reaction Time

Reaction time affects the performance of the LAMP reaction. Short reaction times will not be sufficient to produce detectable amounts of product, leading to undetectability when the amount of DNA template is low. On the contrary, prolonged reaction time will increase product synthesis efficiency, but it can also cause false positives. The false positives when reaction time is long have been shown in previous studies [18,24]. Therefore, a range of different reaction times of 10, 20, 30, 40, 50, and 60 min were investigated. The obtained result indicated that the LAMP product was only observed by agarose gel electrophoresis with reaction times of 50 and 60 min. No false positive was observed with a long reaction time of 60 min (Figure 10).

The summary of optimal conditions for the LAMP reaction in this study is shown in Table 3.

### 3.3. Detection Method

Because the LAMP reaction can produce a large amount of amplified product, in addition to the standard electrophoretic analysis, the results can also be evaluated by turbidity measurement or colorimetric visual inspection using a specific dye. In previous studies on LAMP for *N. meningiditidis*, the product was identified by adding dye after the reaction had ended [17,18,19]. In this study, we used a color indicator that changes color based on the change in pH during the reaction. As a result, the outcome can be observed while the reaction is taking place, without the need to open the tube cap after the reaction is complete (Figure 11). This method minimizes the risk of false positives caused by contamination from amplified LAMP products.

### 3.4. Properties of the LAMP Reaction

#### 3.4.1. Detectability of Different Genotypes and Cross-Reactivity

The *N. meningitidis* LAMP primer set designed in this study was tested with three different genotypes: A, B, and C. Amplification products were observed for all three genotypes on agarose gel electrophoresis (Figure 12). The obtained result indicated that the newly designed primer set can detect three dominant strains of *N. meningitidis*. This primer set was tested against reference DNA templates from *Haemophilus influenzae* type b ATCC 10211 (Hi), *Streptococcus pneumoniae* ATCC 49619 (Sp), *Neisseria lactamica* ATCC 23970, *Staphylococcus aureus* ATCC 29213 (Sa), and *Mycobacterium tuberculosis* strain. None of the reference strains resulted in the amplification product (Figure 13).

#### 3.4.2. Detection Limit of the LAMP Reaction

To determine the limit of detection for *N. meningitidis*, a serious dilution of DNA templates extracted from bacterial culture (10^6^ CFU/mL) from three different genotypes of A, B, and C strains was prepared. The obtained result showed that the LAMP reaction using the new primer set could detect *N. meningitidis* with the limit of 10^2^ CFU/mL (Figure 14).

#### 3.4.3. Sensitivity and Specificity

To determine sensitivity and specificity, the LAMP reaction was tested with 30 *N. meningitidis* positive clinical samples and 30 *N. meningitidis* negative clinical samples. The obtained result showed that the LAMP reaction has a sensitivity and specificity of 100% (95% CI: 88.7–100.0%) (Figure 15 and Figure 16).

## 4. Discussion

Meningococcal meningitis caused by *Neisseria meningitidis* is a serious illness that can significantly impact patients’ health if not detected and treated promptly. Therefore, the development of a fast, inexpensive, and effective diagnostic assay for this disease is essential, and LAMP has emerged as the most optimal method. With the advantage of being fast and not requiring as much complex equipment as PCR, LAMP is a useful tool for field diagnosis of *N. meningitidis*.

We focused on designing a LAMP reaction that can detect the *ctrA* gene, considered a conserved gene often targeted as the key factor in PCR reactions for *N. meningitides* diagnostic. The design of primers is crucial for the success of a LAMP reaction. During our research, we chose the primer set of McKenna et al. [17] because it was the first study to apply LAMP for the detection of *N. meningitidis*. However, false positives were observed when using this primer set. Since this publication was ten years before our study, this primer set may no longer be suitable for the current strains. Therefore, our research group decided to design a new primer set based on the conserved region of the *ctrA* gene.

The optimization of the LAMP reaction involved selecting suitable values for each factor affecting the reaction results. The first factor considered was the primers. Since the reaction used three different primer sets, their concentrations must be in an optimal ratio to prevent reaction inhibition. The three most effective values were determined to be 1.5 µ each of FIB/BIB, 0.2 µM each of F3/B3, and 0.8 µM each of FL/BL. Ions Mg^2+^ play a crucial role in the LAMP reaction. They act as a cofactor for the DNA polymerase enzyme and facilitate the formation of primer/template DNA complexes by stabilizing the negative charges on their phosphate backbones. However, high Mg^2+^ concentrations can also inhibit the LAMP reaction. The optimal MgSO_4_ concentration was determined to be 4 mM. When 10 mM and 12 mM MgSO_4_ were added, no bands were observed on the electrophorestic gel. Adding 8 mM MgSO_4_ resulted in a lower intensity band. For the betaine condition, although the LAMP reaction could still be effective without the addition of betaine, a concentration of 1.2 M was still chosen as betaine is a suitable chemical for avoiding background noise and false-positive results. Melting temperature (T_m_) is a crucial factor affecting the accuracy and efficiency of the LAMP reaction. During temperature optimization, our group observed that the product amplification efficiency remained consistent across various temperatures ranging from 60 °C to 65 °C. No false positives were detected, even at the lowest temperature of 60 °C, so we chose 65 °C as the optimal temperature. While primer/template DNA binding may be more challenging at higher temperatures, this also leads to enhanced reaction specificity and sensitivity, preventing cross-reaction.

Reading LAMP reaction results using electrophoresis is impractical for commercial LAMP kits. Therefore, selecting a result-analysis method suitable for point-of-care testing is important. Our team chose the pH-sensitive dye, which has several advantages. Firstly, it does not inhibit the LAMP reaction at an appropriate concentration. Secondly, the color contrast between positive and negative results is clear and easy to understand, even for non-experts. Thirdly, it saves result processing time, as the color change indicates the presence or absence of the target agent faster than gel electrophoresis. When we performed optimized LAMP reactions with the pH-sensitive dye, the reaction mixes of the positive sample exhibited a clear color change within just 40 min. This remarkably short and rapid turnaround time made our LAMP method suitable for point-of-care testing.

The LAMP primer set designed by our team showed coverage for all three genotypes, A, B, and C, which are the three most common serogroups in the world. The remaining serogroups have not been tested yet, as most cases in Vietnam are currently caused by serogroup B. The LOD of the LAMP reaction is 10^2^ CFU/mL, with both sensitivity and specificity reaching 100%. The LAMP reaction developed by our team showed a significantly lower detectivity compared to the antigen test (10^5^ CFU/mL) [25].

## 5. Conclusions

In conclusion, a novel LAMP primer set targeting the conserved region of the *ctrA* gene of *Neisseria meningitidis* has been designed, and optimal conditions for the LAMP reaction using these primers have been established. However, further evaluation with a larger number of clinical samples is necessary to validate the clinical sensitivity, clinical specificity, and predictive values of the assay.

## Figures and Tables

**Figure 1 cimb-47-00467-f001:**
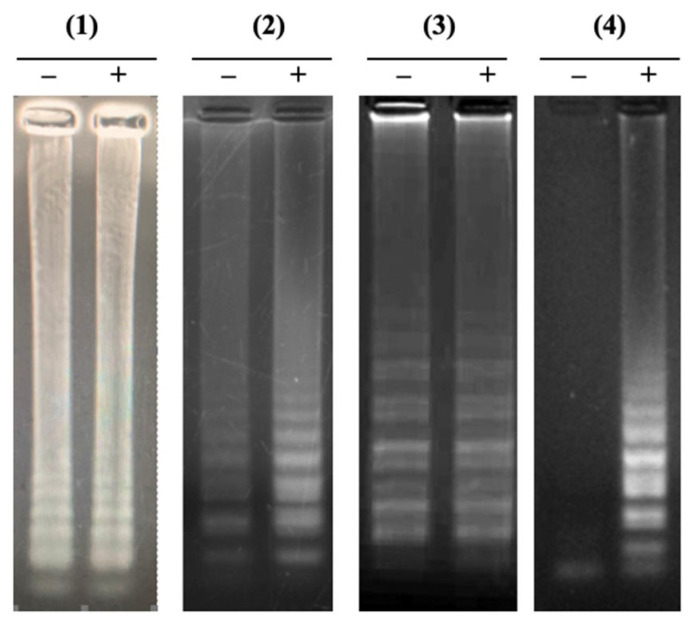
Electrophoretic analysis of LAMP products on 2% agarose gel. The LAMP products showed a multiple-band pattern on the electrophoretic agarose gel. Lanes (–), negative control; Lanes (+), positive control. Panel (**1**). LAMP reaction using the published primers; Panels (**2**–**4**). LAMP reactions using novel designed primers.

**Figure 2 cimb-47-00467-f002:**
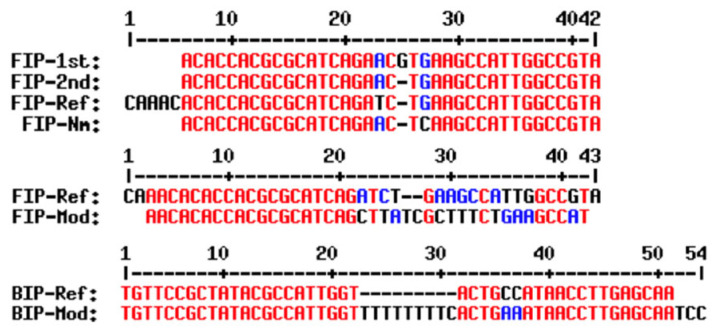
LAMP primers for detecting *N. meningitidis*. FIP-1st, FIP-2nd, and FIP-Nm are FIP primers from [18]; FIP-Ref and BIP-Ref are FIP/BIP primers from [17]; and FIP-Mod/BIP-Mod are FIP/BIP primers designed in this study.

**Figure 3 cimb-47-00467-f003:**
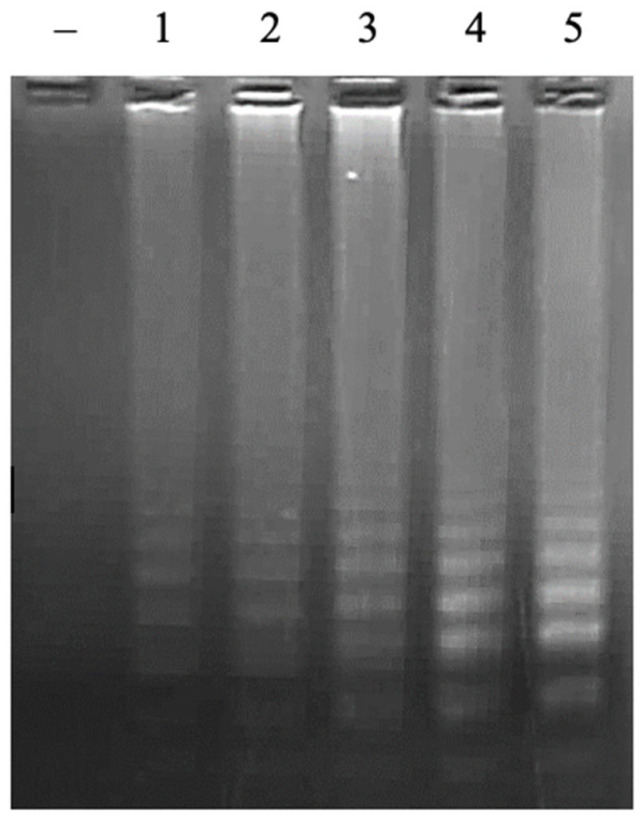
Agarose electrophoretic pattern of LAMP products with different concentrations of FIP/BIP primers. Lane (–), negative control; Lanes 1–5, LAMP products from reactions with different concentrations of FIP/BIP of 0.25, 0.5, 1.0, 1.5, and 2.0 μM, respectively.

**Figure 4 cimb-47-00467-f004:**
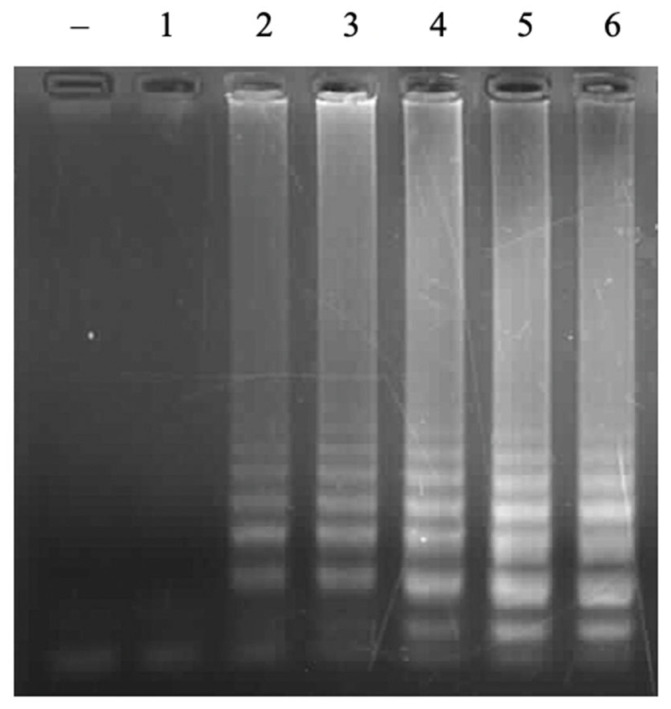
Agarose electrophoretic pattern of LAMP products with different concentrations of loop primers (FL/BL). Lane (–), negative control; Lanes 1–6, LAMP products from reactions with different concentrations of FL/BL of 0, 0.1, 0.2, 0.4, 0.8, and 1.2 μM, respectively.

**Figure 5 cimb-47-00467-f005:**
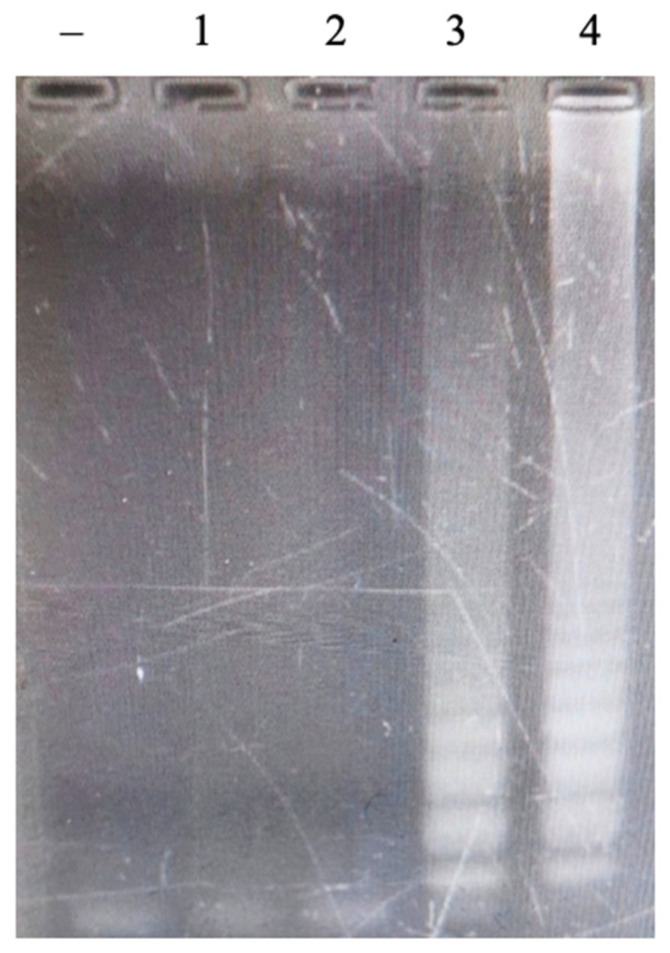
Agarose electrophoretic pattern of LAMP products with different concentrations of dNTPs. Lane (–), negative control; Lanes 1–4, LAMP products from reactions with different concentrations of dNTPs of 0.1, 0.5, 1.0, and 1.5 mM, respectively.

**Figure 6 cimb-47-00467-f006:**
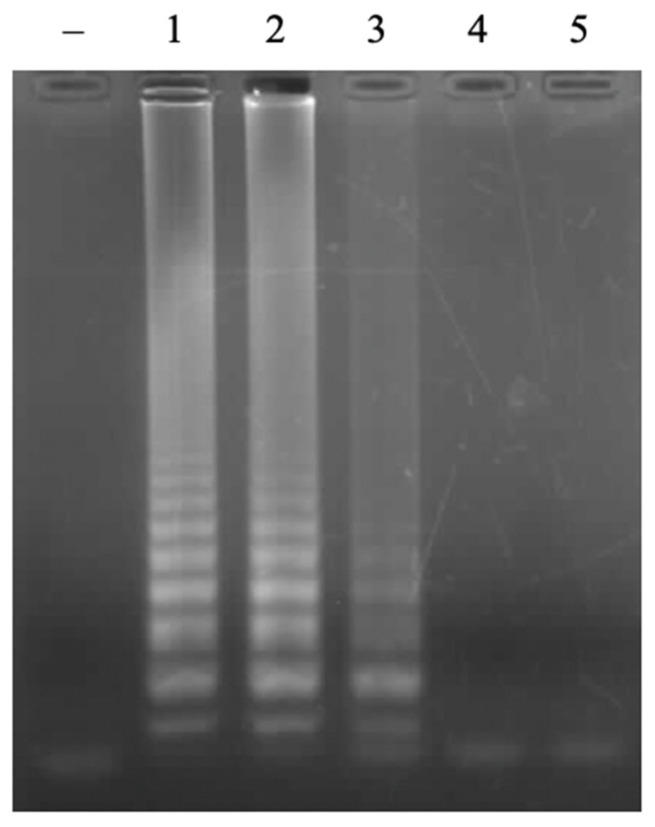
Agarose electrophoretic pattern of LAMP products with different concentrations of MgSO_4_. Lane (–), negative control; Lanes 1–5, LAMP products from reactions with different concentrations of MgSO_4_ of 4, 6, 8, 10, and 12 mM, respectively.

**Figure 7 cimb-47-00467-f007:**
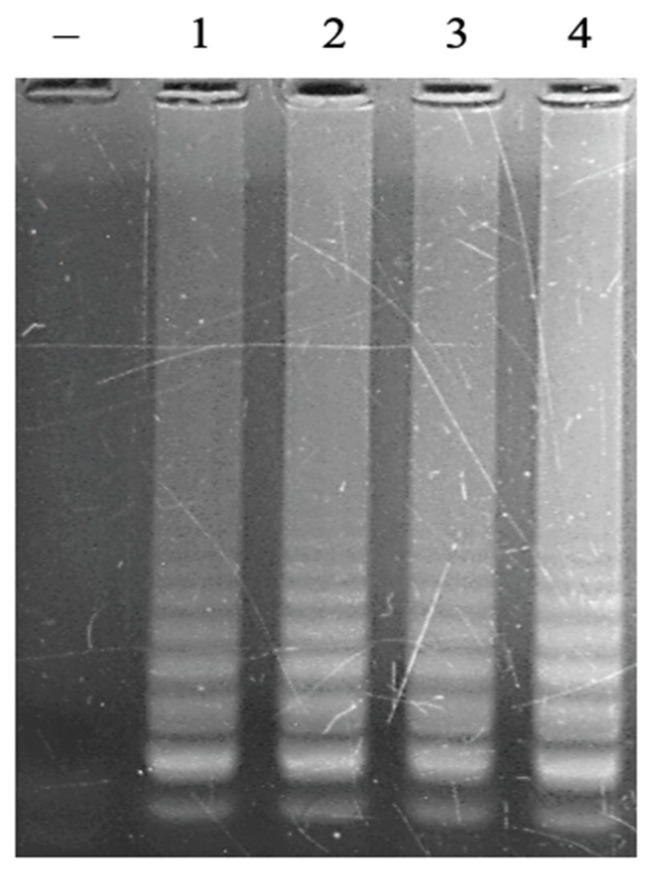
Agarose electrophoretic pattern of LAMP products with different concentrations of betaine. Lane (–), negative control; Lanes 1–4, LAMP products from reactions with different concentrations of betaine of 0, 0.4, 0.8, and 1.2 M, respectively.

**Figure 8 cimb-47-00467-f008:**
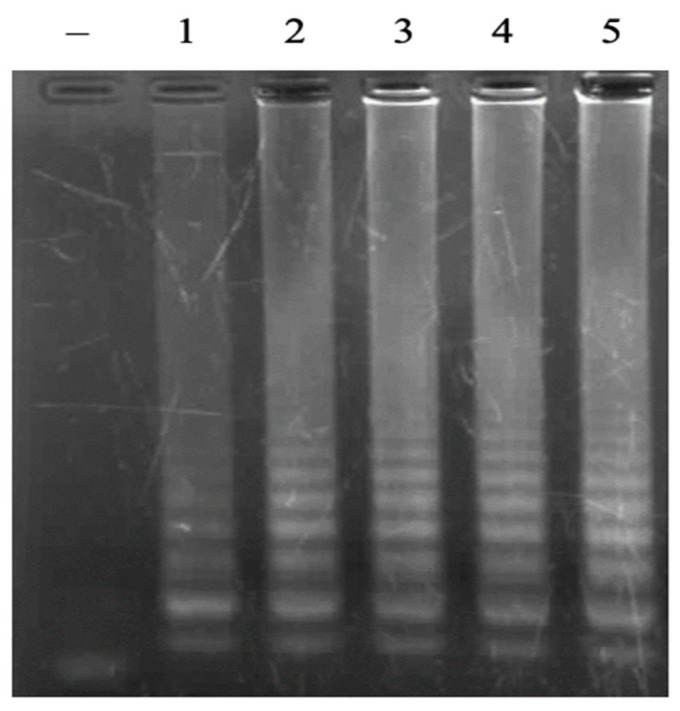
Agarose electrophoretic pattern of LAMP products with different amounts of Bst DNA polymerase. Lane (–), negative control; Lanes 1–5, LAMP products from reactions with different amounts of Bst DNA polymerase of 2, 4, 6, 8, and 10 IU/reaction, respectively.

**Figure 9 cimb-47-00467-f009:**
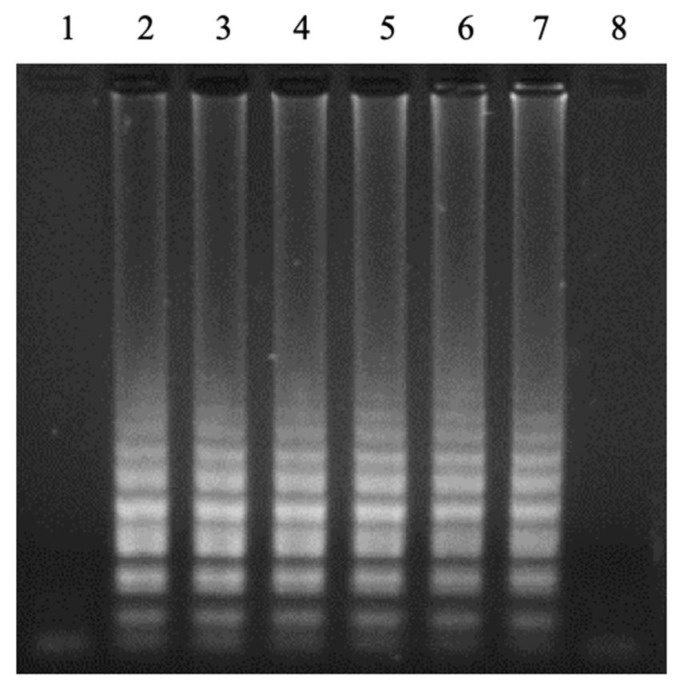
Agarose electrophoretic pattern of LAMP products with reactions at the different temperatures. Lanes 1 and 8, negative control at 60 °C and 65 °C, respectively; Lanes 2–7, LAMP products from reactions at the different temperatures of 60, 61, 62, 63, 64, and 65 °C, respectively.

**Figure 10 cimb-47-00467-f010:**
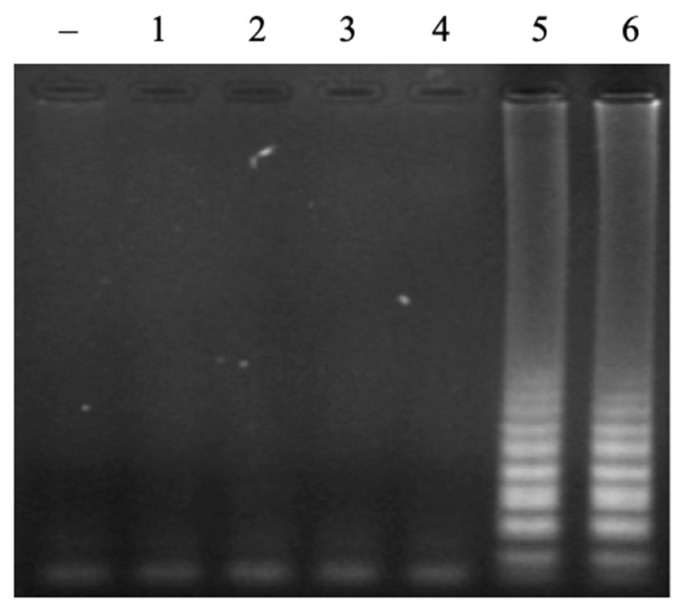
Agarose electrophoretic pattern of LAMP products with different reaction times. Lane (–), negative control; Lanes 1–6, LAMP products from reactions with the different reaction times of 10, 20, 30, 40, 50, and 60 min, respectively.

**Figure 11 cimb-47-00467-f011:**
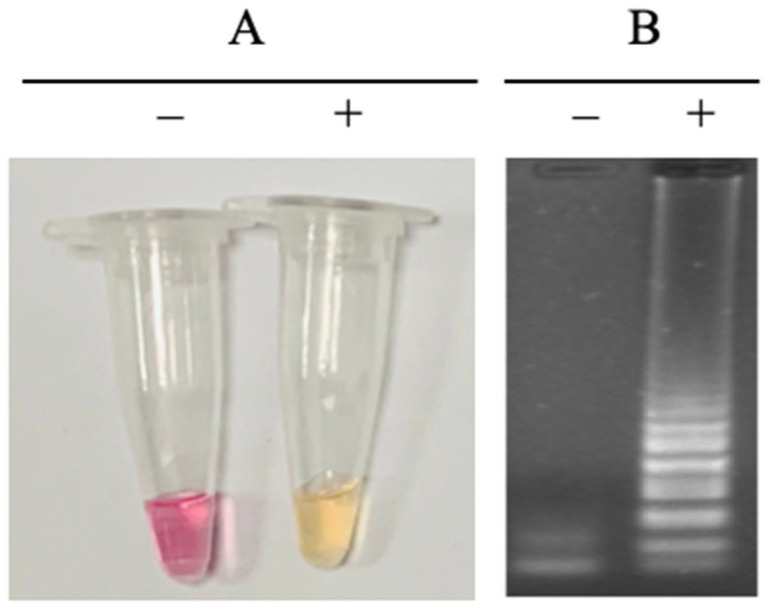
Visual inspection of LAMP reaction (**A**) and agarose electrophoretic pattern of LAMP products (**B**). (−), negative control; (+), positive control.

**Figure 12 cimb-47-00467-f012:**
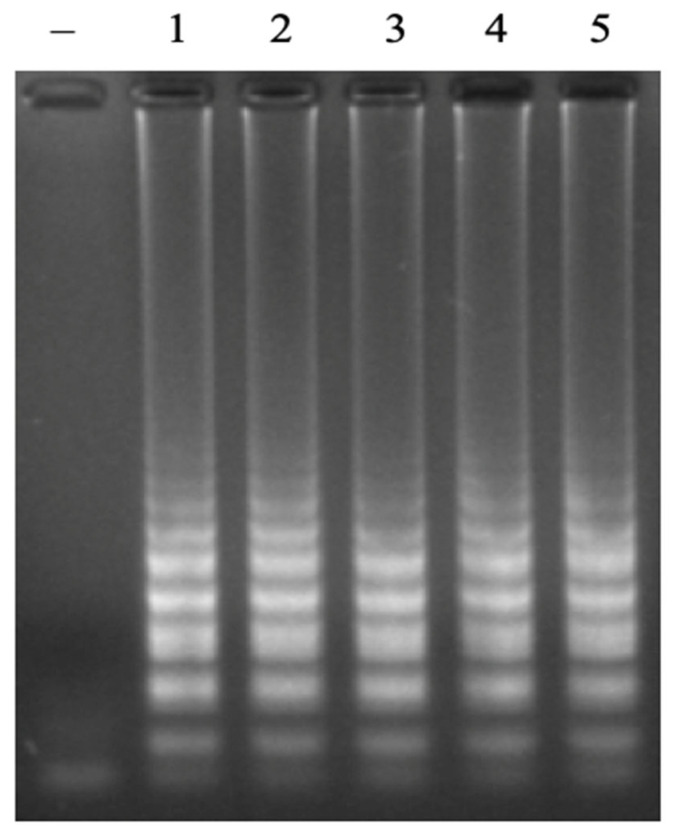
Agarose electrophoretic pattern of LAMP products with different genotypes of *N. meningitidis*. Lane (–), negative control; Lanes 1–5, LAMP products from reactions with the different genotypes of A, B, B, C, and C, respectively.

**Figure 13 cimb-47-00467-f013:**
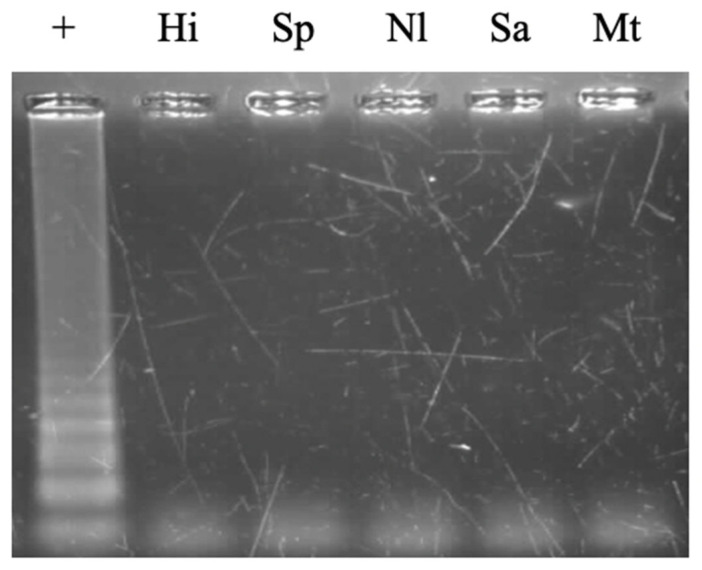
Cross-reactivity evaluation of LAMP reaction with reference strains: *Haemophilus influenzae* type b ATCC 10211 (Hi), *Streptococcus pneumoniae* ATCC 49619 (Sp), *Neisseria lactamica* ATCC 23970 (Nl), *Staphylococcus aureus* ATCC 29213 (Sa), and *Mycobacterium tuberculosis* (Mt); Lane (+), positive control (*N. meningiditis*).

**Figure 14 cimb-47-00467-f014:**
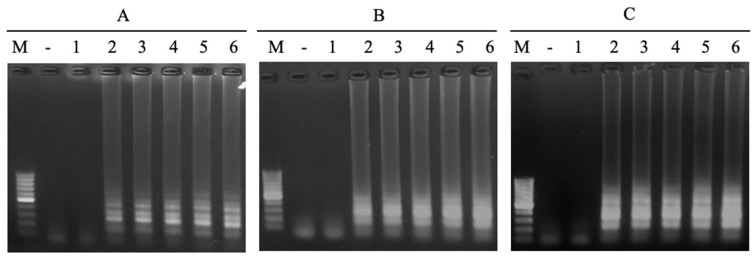
The agarose electrophoretic pattern of the LAMP product using different concentrations of the DNA template from genotypes (**A**), (**B**), and (**C**), respectively. Lane M, DNA marker; Lane (–), negative control; Lanes 1–6, different concentrations of the DNA template (10^1^, 10^2^, 10^3^, 10^4^, 10^5^, 10^6^ CFU/mL).

**Figure 15 cimb-47-00467-f015:**
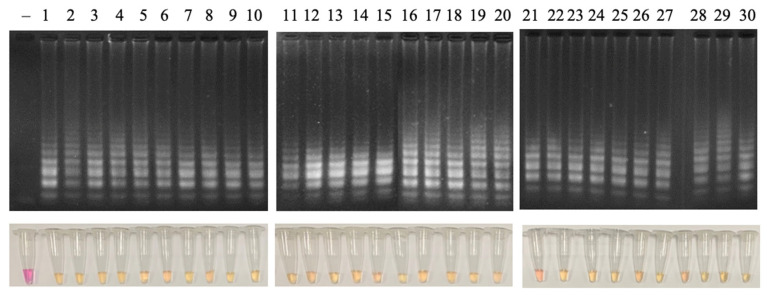
Agarose electrophoretic pattern (**upper layer**) and visual inspection of LAMP products (**lower layer**) of the LAMP reaction with 30 different clinical samples. Lane (−), negative control; Lanes 1–30, clinical samples (positive for *N. meningitidis*).

**Figure 16 cimb-47-00467-f016:**
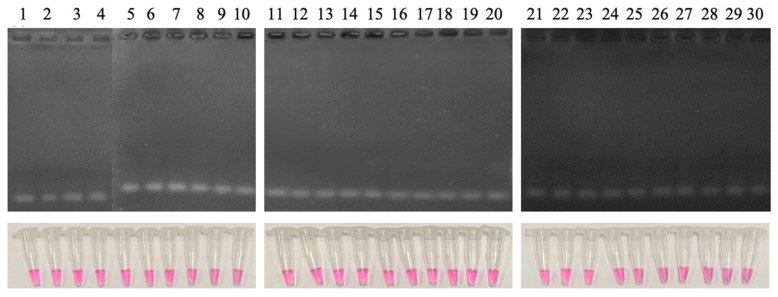
Agarose electrophoretic pattern (**upper layer**) and visual inspection of LAMP products (**lower layer**) of LAMP reaction with 30 different clinical samples. Lanes 1–30, clinical samples (negative for *N. meningitidis*).

**Table 1 cimb-47-00467-t001:** LAMP primer sets for detection of *N. meningitidis.* The first primer set is from a previous study [17]; the second primer set is modified from the first primer set; and the third and fourth sets are newly designed.

Primer Set	Primer Name	Sequence	Tm (°C)	GC (%)
(1)	FIP-1	CAAACACACCACGCGCATCAGATCTGAAGCCATTGGCCGTA	69.7	53.7
	BIP-1	TGTTCCGCTATACGCCATTGGTACTGCCATAACCTTGAGCAA	67.7	47.6
	FL-1	CGATCTTGCAAACCGCCC	57.3	61.1
	BL-1	GCAGAACGTCAGGATAAATGGA	54.9	45.5
	F3-1	AGC(C/T)AGAGGCTTATCGCTT	56.4	52.6
	B3-1	ATACCGTTGGAATCTCTGCC	54.6	50.0
(2)	FIP-2	ACGCTCATCAGAACGGCGATCATCGCTTTCTGAAGCCA	68.9	52.6
	BIP-2	TGTTCCGCTATACGCCATTGGTACTGCCATAACCTTGAGCAA	67.7	47.6
	FL-2	CAAACCGCCCATACGGCC	59.7	66.7
	BL-2	GCAGAACGTCAGGATAAATGGA	54.9	45.5
	F3-2	GGTTTTTCAACCAGAGGCTT	53.5	45.0
	B3-2	ATACCGTTGGAATCTCTGCC	54.6	50.0
(3)	FIP-3	GGCCATTTTTTTCAGGCGGCCTTGGCGATATTTCGGTGGTC	69.4	53.7
	BIP-3	CAAGTGATGGTGCG(C/T)TTGGTGCA(G/A)CGGCATACGCA CACTA	71.1	57.5
	FL-3	ACCTGACCAGGCGTTTTACC	57.6	55.0
	BL-3	CGGTGATTCGTGCTGGGAA	57.9	57.9
	F3-3	ACGTGGTACGGTTTCTGTG	55	52.6
	B3-3	CCACCGCATCCAACACAC	57	61.1
(4)	FIP-4	AACACACCACGCGCATCAGCTTATCGCTTTCTGAAGCCAT	68.8	50.0
	BIP-4	TGTTCCGCTATACGCCATTGGTTTTTTTTTCACTGAAATAA CCTTGAGCAATCC	66.7	38.9
	FL-4	GATCTTGCAAACCGCCCATA	55.7	50.0
	BL-4	CGGCAGAACGTCAGGATAA	54.6	52.6
	F3-4	GGTGGGGAGAACACAAGAAAT	55.1	47.6
	B3-4	AATGCGCATCAGCCATATTCA	61.4	42.9

**Table 2 cimb-47-00467-t002:** Conserved regions of the *ctrA* gene used for primer design.

	Sequence
Conserved region (1)	ACGTGGTACGGTTTCTGTGCCGTTTGTTGGCGATATTTCGGTGGTCGGTAAAACGCCTGGTCAGGTTCAGGAAATTATTAAAGGCCGCCTGAAAAAAATGGCCAATCAGCCGCAAGTGATGGTGCGCTTGGTGCAGAATAATGCGGCAAATGTATCGGTGATTCGCGCAGGCAATAGTGTGCGTATGCCGTTGACGGCAGCCGGTGAGCGTGTGTTGGATGCGGTGG
Conserved region (2)	GGTGGGGAGAACACAAGAAATCGGTTTTTCAGCTAGAGGCTTATCGCTTTCTGAAGCCATTGGCCGTATGGGCGGTTTGCAAGATCGCCGTTCTGATACGCGTGGTGTGTTTGTGTTCCGCTATACGCCATTGGTGGAATTGCCGGCAGAACGTCAGGATAAATGGATTGCTCAAGGTTATGGCAGTGAGGCAGAGATTCCAACGGTATATCGTGTGAATATGGCTGATGCGCATT

**Table 3 cimb-47-00467-t003:** A summary of optimized LAMP reaction conditions.

LAMP Reaction Components	Concentration/Conditions
FIP/BIP primers	1.5 µM
FL/BL primers	0.8 µM
dNTPs	1.5 mM
MgSO_4_	4 mM
Betaine	1.2 M
Bst DNA polymerase	6 IU
Reaction temperature	65 °C
Reaction time	50 min

## Data Availability

Data is contained within the article.

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
