# Peer review of "Design of New Primer Sets for the Development of a Loop-Mediated Isothermal Amplification for Rapid Detection of *Neisseria meningitidis"

_cimb, 2025, doi:10.3390/cimb47060467_

Round 1

Reviewer 1 Report

Comments and Suggestions for Authors

The study aims to design and optimize a novel LAMP primer set targeting the ctrA gene for the rapid, low cost, sensitive and specific detection of Neisseria meningitidis. The manuscript addresses an important diagnostic gap and provides a LAMP assay that is optimized and provides real-time visual readout.

Based on the results, the authors conclude that this LAMP assay is rapid (40 min), sensitive (10² cfu/ml), and 100% specific across tested samples, by conducting primer specificity (no false positives) and sensitivity testing (no amplification with S. pneumoniae, S. aureus, H. influenzae), and clinical sample evaluation (30 positive and 30 negative).

The references are relevant and up to date.

However, I would like to make a few suggestions regarding the methodology improvement:

  • The manuscript does not provide data on cross-reactivity with closely related Neisseria species ( gonorrhoeae or N. lactamica).
  • There is no mention of an internal amplification control to detect inhibitors or monitor reaction efficiency.
  • There is no mention of intra-assay tests for repeatability, inter-assay tests for reproducibility or inter-laboratory comparisons for external quality assessment.
  • There is no mention of sequencing LAMP amplicons as part of the specificity validation.
  • The composition on Negative control (ddH2O)?
  • A comparison against gold-standard culture or PCR would add to the effectiveness of the suggested LAMP assay.

Regarding the statistical analysis:

  • Reporting sensitivity and specificity as point estimates (100%) without 95% CIs gives a false sense of precision, therefore I would suggest reporting exact 95% CIs for both sensitivity and specificity.
  • The manuscript does not explain why 30 positive and 30 negative samples were chosen or whether this number is sufficient to demonstrate the experiment with acceptable precision. Therefore, I would suggest including a sample-size calculation based on expected sensitivity/specificity, and alpha level.
  • There is no statistical test comparing the LAMP results directly with the compared method (PCR) on the same samples.
  • The authors state the statistical analysis was performed using a tool in https://www.socscistatistics.com, but do not specify which statistical module or its parameters.

Additional comments:

  • A summary table of optimized reaction conditions (primers, concentrations, temperatures, time) would help readers for a quick reference.
  • Regarding “Table 1. LAMP primer sets for detection of meningitidis” and “Table 2. Conserved regions used for primer design” (pages 4-5) more detailed legends should be provided.
  • In the legend of Figure 1 (page 5) there is the sentence “Table 1: Statistical analysis of the testing results of the generated LFA test strip with mork clinical samples.”

Overall, I would suggest reconsidering after major revisions.

Reviewer 2 Report

Comments and Suggestions for Authors

The work describes a relevant study on the development and optimization of a LAMP assay for Neisseria meningitidis detection. The work reported is timely, and the experimental work appears thorough. However, several issues related to clarity, writing quality, organization, and scientific framing need attention before the manuscript is suitable for publication.

General thoughts

  • The manuscript has to be edited appropriately, that is, language editing to address grammar and phrasing inconsistencies.
  • Figures and tables should be referenced more clearly and integrated into the narrative.

Line-by-Line Comments and Suggested Revisions

Lines 16–17 (Abstract)
Comment: The final sentence of the abstract may be better worded.
Suggested revision: "This result suggests the assay could be a promising tool for developing a rapid N. meningitidis detection method suitable for use in Vietnam and other developing countries."

Lines 21–26
Comment: There is a typo in "sensitiviy."
Suggested revision: Correct to "sensitivity."

Line 32

Comment: Remove “Nowadays”

Lines 32–34

Comment: “The capsular polysaccharide…………. prevalent worldwide [2]”

Suggested revision: This sentence is unclear and needs rephrasing

Lines 43–45
Comment: The sentence needs editing; it is unclear. What do you mean by “the specific disruption information of N. meningitidis serogroup is limited and needs further studies [6]”

Lines 48–49
Comment: This part of the sentence needs editing: “antibiotics before or incorrect culturing method”.

Suggested revision: sentence needs correcting

Lines 60

Comment: Remove “the” before “the PCR” to read “to PCR” instead.

Lines 60–61
Comment: The phrasing "this method does not require complicated equipment or machines" is awkward.
Suggested revision: Consider rephrasing to "this method operates without the need for complex instrumentation."

Line 76
Comment: The sentence "Our LAMP assay is an effective and suitable diagnostic method..." is too conclusive for this section.
Suggested revision: Rephrase to "We propose that the LAMP assay developed here could serve as an effective and suitable diagnostic method..."

Lines 144–148
Comment: This section would benefit from better clarity on how the primers were evaluated.
Suggested revision: Include a sentence summarizing the rationale for choosing the primer set.

Figure Captions (e.g., Figures 1–16)
Comment: Several captions are missing clarity or have formatting issues. For example, "panel (1)" should be "Panel 1" and "mork" should be "mock." Suggested revision: Review all figure captions for spelling, capitalization, and clarity.

Lines 387–392
Comment: Incorrect units are given for MgSO4 (e.g., "4 M" instead of "4 mM").
Suggested revision: Revise all instances where molarity is misstated.

Line 417–419 (Discussion)
Comment: The statement that the LAMP assay "demonstrates significantly higher sensitivity and specificity" than PCR is unsubstantiated.
Suggested revision: Either provide comparative PCR data or soften the language.

Lines 421–425 (Conclusion)
Comment: The conclusion is repetitive and overly optimistic.
Suggested revision: Streamline the conclusion and reflect any study limitations.

Minor Suggestions Throughout:

  • Standardize terminology (e.g., LAMP reaction vs. LAMP assay).
  • Double-check spelling of "meningitidi" — should be "meningitidis."
  • Ensure consistent use of µM, mM, IU, and other units.

Recommendation: Major revisions. The work is promising and could make a good contribution once these issues are addressed.

Comments on the Quality of English Language

The manuscript has to be edited appropriately, that is, language editing to address grammar and phrasing inconsistencies.

Reviewer 3 Report

Comments and Suggestions for Authors

The researcher of this study developed the new primer set, which is highly sensitive and specific to detect N. meningitidis using LAMP assay. The results for the optimization of the different factors affecting the assay is clearly presented. 

I have some minor comments: 

  1. In the abstract, the term "dangerous bacterial agent" is understandable but is not a standard scientific term.
  2. In introduction, it would be good to specify why new primer set has to be designed, as there were already designed primer sets for LAMP assay, and adding the principle of the assay.
  3. To detect limit of detection of bacterial CFUs, it is mentioned about the "serious dilution" of template DNA. Clarify how the limit of detection was calculated? Is DNA dilutions correlated to the CFUs count?

Round 2

Reviewer 1 Report

Comments and Suggestions for Authors

The authors have taken into consideration the reviewers' comments and revised the manuscript accordingly.

I would recommend accept in present form.